# Morphological Diversity of Different Male Morphotypes of Giant Freshwater Prawn *Macrobrachium rosenbergii* (De Man, 1879)

**Salifu Ibrahim** [1], **Zhenxiao Zhong** [1], **Xuan Lan** [1], **Jinping Luo** [1], **Qiongying Tang** [1,*], **Zhenglong Xia** [2], **Shaokui Yi** [1] and **Guoliang Yang** [1,2,*]

1   Zhejiang Province Key Laboratory of Aquatic Resources Conservation and Development, College of Life Sciences, Huzhou University, Huzhou 313000, China; ibsuala11@gmail.com (S.I.)
2   Jiangsu Shufeng Prawn Breeding Co., Ltd., Gaoyou 225654, China
*   Correspondence: tangqy@zjhu.edu.cn (Q.T.); ygl0572@163.com (G.Y.)

**Abstract:** The giant freshwater prawn (GFP), *Macrobrachium rosenbergii*, is one of the largest palaemonids in the world, found in tropical marine, estuarine, and freshwaters, and is among the most commercially cultured crustaceans. According to research, mature males usually develop differences in cheliped morphology, growth characteristics, and agonistic behavior. The identification of such morphotypes is critical for effectively managing and handling prawns. The present study aimed to describe the GFP male population structure in culture ponds (the Yangtze River delta, China). Sixteen morphometric traits and four weight data were measured for each four male morphotype. Principal component and clustering analyses were conducted to investigate the morphological variation among the four morphotypes. The study of relative growth was also employed to estimate the growth patterns of body structures (dependent variables) in relation to the carapace length (independent variable). A detailed description of the cheliped's macroscopic characteristics that differed among morphotypes was provided, which corroborated with previous studies of the species. The four morphotypes were statistically different regarding the cheliped morphology, size, and morphometric relationships and equations, indicating a considerable variation in growth among the four male morphotypes. The present results contribute to a clear understanding of the population biology of GFP and support future management and broodstock selection activities.

**Keywords:** *Macrobrachium rosenbergii*; morphological diversity; chelipeds; morphotypes; relative growth

## 1. Introduction

The giant freshwater prawn (GFP) *Macrobrachium rosenbergii* (De Man, 1879), is a freshwater decapod of the most significant economic importance in China and other Southeast Asian countries such as India, Thailand, Vietnam, and Bangladesh. It is widely cultivated for its great value as a food source, good economic returns, and excellent disease resistance [1]. GFP farming has increasingly become an important area of the aquaculture industry in China, ever since it was first introduced from Japan in 1976 [2], accounting for about 50–60% of total global production in recent years [3]. Despite this outstanding growth, the development of the GFP farming industry is somewhat overshadowed by several critical issues, such as slow growth rate, size variation at harvest, disease, and deterioration of the pond environment [4].

In most *Macrobrachium* species, individuals of the same sex, usually males, exhibit differential growth patterns [5], termed heterogeneous individual growth (HIG), giving rise to more than one morphotypes that differ in size, morphology, physiology, and behavior [6]. Male morphological differentiation was recorded in populations of *M. amazonicum* Heller, 1862 [7], *M. macrobrachion* Herklots, 1851 [8], *M. grandimanus* Randall, 1840 [5], *M. idella*

Hilgendorf, 1898 [9], *M. brasiliense* [10], and in some species of the genus *Cryphiops* such as *Cryphiops caementarius* Molina, 1782 [11]. Recognition of morphological diversity in the species is critical for developing conservation strategies for grow-out ponds to maximize yield and profitability for the sector, particularly in the selection and management of broodstock.

In GFP, the morphotypes' development is an irreversible sequential process resulting in three main morphotypes [12] and several intermediate forms [13]. These morphological patterns indicate different stages of ontogenetic development of the male maturation process [14], from the small male (SM) via the orange claw (OC) to the blue claw (BC) morphotype. The SM morphotypes are small prawns with fine translucent claws and a very slow growth rate. They grow and metamorphosize into OC morphotypes, with large orange claws on their major chelipeds. BC males are large and have very long blue claws [15]. The three morphotypes result from the same age group, forming a complex social hierarchy. Their morphological characteristics, reproductive activities, social status, growth rate, etc., vary greatly [16]. Aside from the three most discussed morphotypes (SM, OC, and BC), other morphotypes often found in ponds include individuals who have lost their chelipeds due to fighting (no claw males) and senescent individuals (old blue claw males) [16]. But only a few studies mentioned these morphotypes. In this study, we identified and added old, blue claw males (OBC) to the three main male morphotypes, thus using four morphotypes as our experimental materials to analyze the extent of variability and relationship among the GFP male morphotypes for the first time in mainland China, allowing for a more detailed and complete GFP male population structure. Here, we identify the structures that best differentiate the various morphotypes, and describe the external morphology and allometric relationships among morphological variables. We also describe the intraspecific variation in the major cheliped morphology among the male morphotypes. This knowledge may be useful for future research on the biology and culture of this species, providing cues for broodstock selection and managing the heterogeneous growth in grow-out ponds to maximize production.

## 2. Materials and Methods

### 2.1. Collection of Specimens and Sampling

In the autumn of 2021, *M. rosenbergii* samples were collected from Jiangsu Shufeng Prawn Breeding Co. Ltd., Gaoyou, in Jiangsu province, China. Male prawns were collected from a single-age population (i.e., a single family) of about 140 days of growth in a 0.2 ha earthen pond at a depth of 2–2.2 m and a density of 10–12 prawns/m$^2$. The samples were kept in plastic buckets equipped with freshwater in the culture pond, and transported to the laboratory. A total of 215 male prawns, including four morphotypes with undamaged appendages, were used for the morphological analysis.

### 2.2. Identification of Morphotypes

Identification and classification of morphotypes were performed according to the keys and methods proposed by Kuris et al. (1987) [17]. Because OBC males and BC males can occasionally be mistaken, we took great care to distinguish between the two morphotypes. The OBC morphotype is characterized by relatively smaller abdominal length in relation to carapace length and major cheliped length [15]. On that basis, the selected 215 prawns were divided into four morphotype groups: small males (SM) = 62, blue claw (BC) = 40, orange claw (OC) = 62, and old blue claw (OBC) = 51.

### 2.3. Morphometric Study

After collection, identification, and classification, the following 16 body dimensions and appendage lengths were measured with a digital caliper (0.01 mm) (Figure 1: total length (TL), body length (BL), rostrum length (RL), carapace length (CL), carapace depth (CD), carapace width (CW), abdominal length (AL), abdominal depth (AD), abdominal width (AW), major cheliped length (MCL), telson length (TeL), uropod length (UL), propo-

dus length (PrL), propodus width (PrW), carpus length (CaL) and carpus width (CaW). Four weight measurements were taken to the nearest 0.01 g using an electronic Sartorius balance: total body weight (Bw), carapace weight (Cw), major cheliped weight (MCw), and abdominal weight (Aw). The morphological characters were described according to Wortham and Maurik (2012) [5] and Kuris et al. (1987) [17].

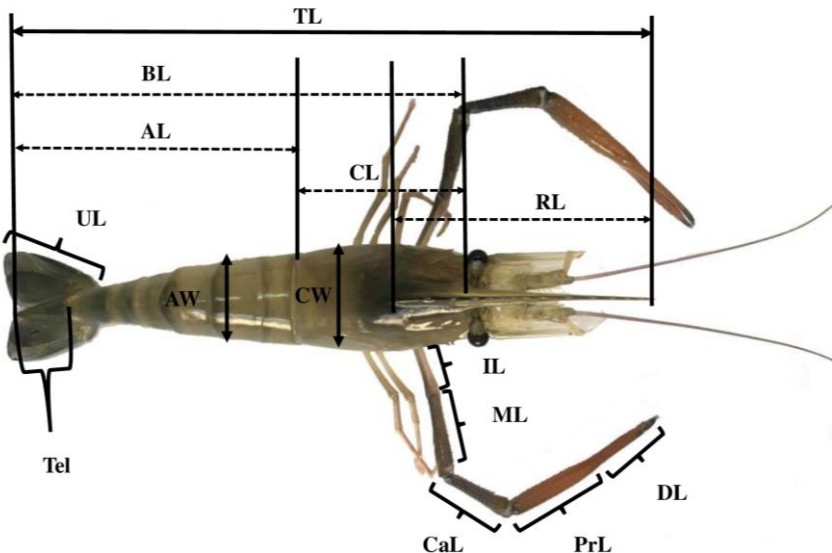

**Figure 1.** Dimensions used for the morphometric analyses of *M. rosenbergii.* TL, total length; AL, abdominal length; AW, abdominal width; BL, body length; RL, rostrum length; CL, carapace length; CW, carapace width; CaL, carpus length; DL, dactylus length; IL, ischium length; ML, merus length; PrL, propodus length; TeL, telson length; UL, uropod length.

After completing the measurements, we investigated the spination and color of the major cheliped [17]. These characteristics were crucial in distinguishing *Macrobrachium* morphotypes [5,7,10,18]. Chelipeds of 15 prawns per group were photographed with a Sony A7riii digital camera (50 megapixels) (Sony, Tokyo, Japan). We then used the photographs to measure 20 spines: the angles (relative to the surface) and heights (distance from the basis to the spine) on each segment, using the Image J 1.44 software tool. The mean and standard deviation of these dimensions of spines were then computed for each group. Turkey HSD, a multiple comparisons test, was used to compare the significant differences among morphotypes ($p < 0.05$).

*2.4. Statistical Analyses*

All statistical analyses were carried out using the SPSS (version 25.0) or MINITAB (Version. 17.0) statistical packages and Excel 2021 (Microsoft Corporation, Redmond, WA, USA). Firstly, the data were subjected to the Shapiro–Wilk test to check data normality, and the casewise diagnostics method was used to check the existence of outliers in the dataset. Thirteen outliers were identified and removed [19]. Variations in the morphometry characteristics among the morphotypes were analyzed using the Kruskal–Wallis non-parametric analysis of variance (ANOVA), checking for the mean, standard deviation, maximum and minimum values of all the variables for each morphotype. The multivariate analyses used in this study were the principal component (PCA) and cluster analyses (CA). The PCA was performed to evaluate the morphometric variation among the four morphotypes and identify variables substantially contributing to that variation [20]. The CA was performed using the non-hierarchical K-means clustering analysis method to categorize morphotypes based on distances, using an algorithm that divides each case into unique clusters before combining them. A dendrogram was used to illustrate the procedure.

The equations representing the relationships between the carapace length (a measurement of body growth) and the other morphometric variables were determined for each

morphotype, and were described by the power function $y = ax^b$ [21], and the curvilinear equation was transformed into the logarithmic equation $\log y = b \times \log x + \log a$, where $y$ represents the length of the dependent variable or a given body structure, $x$ means the carapace length (CL) (independent variable), $a$ is the intercept and $b$ is the slope of the transformed equation and describes the rate of growth of the dependent variable relative to the reference dimension (CL). Carapace length was chosen as the reference dimension in this study because it is the easiest, fastest, and most reliable to measure [17]. The statistical comparison of growth rate was performed in agreement with Kuris et al. (1987) [17] and Moraes-Riodades and Valenti (2004) [7]. Growth of a particular variable was defined as positive allometry when $b > 1.1$, negative allometry when $b < 0.9$, and isometry at $0.9 < b < 1.1$ [10,22]. The adjusted equations for each group were compared by linear multiple regression analyses using SPSS, with the significance level set at $p < 0.001$. Scatter plots were constructed from the equations obtained by the regression analyses of carapace length and body structures.

## 3. Results

### 3.1. The Variance of Morphometric Traits

The descriptive statistical results of all selected morphometric traits of the four male morphotypes are presented in Table 1. Most of the observed morphometric characteristics demonstrated a highly significant difference ($p < 0.05$) among the four morphotypes, except for abdominal length (AL), abdominal width (AW), abdominal depth (AD), carapace length (CL), carapace width (CW), carapace depth (CD), carpus length (CaL), carpus width (CaW), carapace weight (Cw) between BC and OC, and abdominal width (AW), abdominal depth (AD) between BC and OBC, carapace depth (CD) and propodus width (PrW) between OC and OBC. The OC males had the highest mean values, especially in terms of TL, BL, and CL, followed by BC, OBC, and SM. However, the largest individual was found in the BC group (Table 1).

**Table 1.** Mean and standard deviation values of the morphometric variables evaluated in the adult male morphotypes of *M. rosenbergii*.

| Morphometric Variables | Morphotypes | | | | | | | |
|---|---|---|---|---|---|---|---|---|
| | SM (*n* = 62) | | BC (*n* = 40) | | OC (*n* = 62) | | OBC (*n* = 51) | |
| | Mean ± SD | Range | Mean ± SD | Range | Mean ± SD | Range | Mean ± SD | Range |
| Total length | 89.87 ± 10.62 [d] | 72.68–116.68 | 161.36 ± 11.47 [b] | 140.00–186.00 | 170.16 ± 10.78 [a] | 131.00–183.00 | 143.07 ± 16.47 [c] | 117.45–225.95 |
| Body length | 67.32 ± 9.42 [d] | 52.64–108.51 | 123.74 ± 9.16 [b] | 107.02–143.91 | 129.48 ± 8.28 [a] | 96.71–139.30 | 110.15 ± 11.55 [c] | 90.16–157.65 |
| Rostrum length | 33.15 ± 4.74 [d] | 23.17–44.25 | 59.42 ± 5.19 [b] | 43.56–70.23 | 61.66 ± 4.69 [a] | 43.48–69.21 | 52.96 ± 8.14 [c] | 26.78–95.30 |
| Abdominal length | 45.03 ± 5.04 [c] | 36.45–58.61 | 79.73 ± 5.92 [a] | 65.65– 96.42 | 81.02 ± 5.26 [a] | 63.47– 90.10 | 69.91 ± 8.29 [b] | 58.22– 110.24 |
| Abdominal width | 9.80 ± 1.36 [c] | 7.93–13.74 | 20.08 ± 2.30 [ab] | 7.98–54.03 | 21.23 ± 4.84 [a] | 15.18–50.61 | 18.33 ± 5.24 [b] | 14.14–52.9 |
| Abdominal depth | 12.41 ± 1.28 [c] | 9.98–15.17 | 23.61 ± 2.78 [ab] | 12.56–60.88 | 24.87 ± 5.04 [a] | 18.58–57.16 | 21.54 ± 5.14 [b] | 17.74–55.31 |
| Carapace length | 20.71 ± 2.8 [c] | 15.61–27.68 | 43.50 ± 4.35 [a] | 29.59–55.3 | 45.82 ± 3.65 [a] | 33.22–52.9 | 40.17 ± 6.40 [b] | 32.65–78.02 |
| Carapace width | 12.18 ± 1.9 [c] | 9.21–18.58 | 26.73 ± 3.02 [a] | 14.63–33.56 | 28.33 ± 2.53 [a] | 19.26–30.99 | 24.77 ± 5.54 [b] | 19.57–59.7 |
| Carapace depth | 13.95 ± 1.99 [c] | 10.37–18.5 | 32.29 ± 3.02 [a] | 21.25–41.23 | 32.62 ± 2.90 [ab] | 23.73–39.85 | 30.68 ± 7.47 [b] | 24.49–67.78 |
| Major cheliped length | 50.79 ± 8.94 [d] | 36.40–76.89 | 140.46 ± 21.86 [c] | 105.37–275.20 | 159.58 ± 28.64 [b] | 85.00–196.00 | 193.32 ± 30.09 [a] | 151.49–260.00 |
| Propodus length | 10.23 ± 1.96 [d] | 6.71–15.46 | 32.07 ± 7.05 [c] | 16.23–68.20 | 36.61 ± 7.92 [b] | 17.62–50.48 | 48.13 ± 7.73 [a] | 33.20–67.62 |
| Propodus width | 1.90 ± 0.47 [c] | 1.15–3.01 | 6.41 ± 1.18 [b] | 3.67–10.57 | 7.39 ± 1.34 [a] | 3.49–8.37 | 7.05 ± 0.97 [a] | 5.18–9.64 |
| Carpus length | 11.81 ± 2.25 [c] | 8.21–17.38 | 29.28 ± 6.76 [b] | 20.81–66.92 | 34.30 ± 6.77 [b] | 19.10–45.72 | 46.49 ± 10.70 [a] | 32.35–86.77 |
| Carpus width | 1.69 ± 0.45 [c] | 0.97–2.67 | 6.14 ± 1.06 [b] | 4.31–9.72 | 6.65 ± 1.19 [ab] | 3.24–8.41 | 7.06 ± 1.08 [a] | 5.11–9.57 |
| Telson length | 11.18 ± 1.35 [d] | 8.64–14.49 | 20.05 ± 1.45 [b] | 17.28–56.04 | 21.80 ± 4.85 [a] | 16.52–56.54 | 18.33 ± 1.77 [c] | 14.96–23.78 |
| Uropod length | 15.26 ± 1.85 [d] | 11.24–19.60 | 26.48 ± 3.14 [b] | 16.96–65.94 | 28.39 ± 5.37 [a] | 20.88–31.86 | 23.77 ± 2.45 [c] | 18.27–29.53 |
| Major cheliped weight | 0.14 ± 0.17 [d] | 0.01–0.70 | 5.14 ± 2.40 [c] | 2.02–30.71 | 8.98 ± 4.53 [b] | 1.40–11.25 | 11.22 ± 5.49 [a] | 3.18–27.37 |
| Carapace weight | 2.91 ± 1.24 [c] | 1.21–6.67 | 28.29 ± 5.71 [a] | 19.86–41.12 | 29.55 ± 6.03 [a] | 11.12–38.13 | 20.51 ± 6.02 [b] | 11.57–35.67 |
| Abdominal weight | 3.74 ± 1.36 [d] | 1.73–7.88 | 22.47 ± 4.24 [b] | 13.40–35.92 | 25.25 ± 4.03 [a] | 11.42–33.97 | 14.66 ± 4.35 [c] | 8.31–24.83 |
| Wet weight | 7.25 ± 2.86 [d] | 3.20–16.20 | 58.83 ± 10.85 [b] | 40.59–99.60 | 68.52 ± 14.19 [a] | 25.00–79.35 | 48.34 ± 15.14 [c] | 25.70–88.36 |

Note: values in the same row having different superscripts are significantly different ($p < 0.05$). The unit for length and weight is mm and g, respectively.

The principal component analysis of all the morphometric traits extracted two principal components (PC1 and PC2) (Table 2). The highest variance in the total variability was contributed by PC1 (78.302%). The PC2, on the other hand, accounted for an 8.46% variance

in the total variability. According to the loadings of component coefficients obtained for morphometric data, the most influential variables for PC1 included CaW, TL, RL, BL, CL, CW, CD, AL, AD, Cw, and Aw. The highest contributions were from CL (0.978), while the lowest was from Bw (0.141). Variation in PC2 was contributed mainly by CaL, MCw, MCL, PrL, and Bw, with the highest contributions from CaL (0.544). The result of the PCA analysis suggests that these variables could be used to distinguish the male morphotypes of *M. rosenbergii*.

**Table 2.** Principal component analysis of morphological variables in male *M. rosenbergii*, showing factor coordinates based on contributions of morphometric variables, for the first two principal components (PC1, PC2).

| Variables | Components | |
|---|---|---|
| | PC1 | PC2 |
| Body weight (Bw) | 0.141 | 0.398 |
| Major cheliped weight (MCw) | 0.778 | 0.503 |
| Major cheliped length (MCL) | 0.883 | 0.429 |
| Propodus length (PrL) | 0.844 | 0.491 |
| Propodus width (PrW) | 0.834 | 0.22 |
| Carpus length (CaL) | 0.782 | 0.544 |
| Carpus width (CaW) | **0.936** | 0.256 |
| Total length (TL) | **0.965** | −0.174 |
| Rostrum length (RL) | **0.947** | −0.142 |
| Body length (BL) | **0.967** | −0.149 |
| Carapace length (CL) | **0.978** | −0.098 |
| Carapace width (CW) | **0.969** | −0.109 |
| Carapace depth (CD) | **0.943** | −0.041 |
| Abdominal length (AL) | **0.964** | −0.199 |
| Abdominal width (AW) | 0.892 | −0.227 |
| Abdominal depth (AD) | **0.909** | −0.225 |
| Telson length (TeL) | 0.871 | −0.236 |
| Uropod length (UL) | 0.893 | −0.253 |
| Carapace weight (Cw) | **0.941** | −0.149 |
| Abdominal weight (Aw) | **0.908** | −0.256 |
| Cumulative variance explained | 78.302 | 8.46 |
| Eigenvalues | 15.66 | 1.69 |

A hierarchical cluster analysis of the four morphotypes formed three well-defined groups, as revealed by the scatter dendrogram (Figure 2) and the scatterplot derived after plotting PC1 and PC2 (Figure 3). Overall, the morphotypes BC and OC were more similar. Morphotypes SM and OBC were far off from the other groups. The dissimilarities between the morphotypes are further supported by scatterplots derived by the relative growth analysis of the six morphometric variables against carapace length (Figure 4).

*3.2. Relative Growth Analysis*

The relative growth analysis of six key morphometric traits in relation to the independent variable carapace length (CL) is given in Table 3. Scatterplots and equations are shown in Figure 4. From the morphometric relationships and equations, the four morphotypes of *M. rosenbergii* (SM, BC, OC, and OBC) significantly differ in growth patterns of various morphometric characters, demonstrated by the differences in the allometric growth constant. Linear regressions showed that TL vs. CL, AL vs. CL, and RL vs. CL relationships were negatively allometric in all groups, indicating that carapace length grows faster than total length, rostrum length, and abdominal length. The relationship of MCL vs. CL was isometric in BC and OC, positively allometric in larger male OBC and negatively allometric in SM. A similar association was found in CaL vs. CL and PrL vs. CL. Thus, the chelipeds, carpus, and propodus grow faster in the bigger males (BC, OC and OBC) than in the small males. Regarding the SM males, the growth of almost all the structures was negatively allometric.

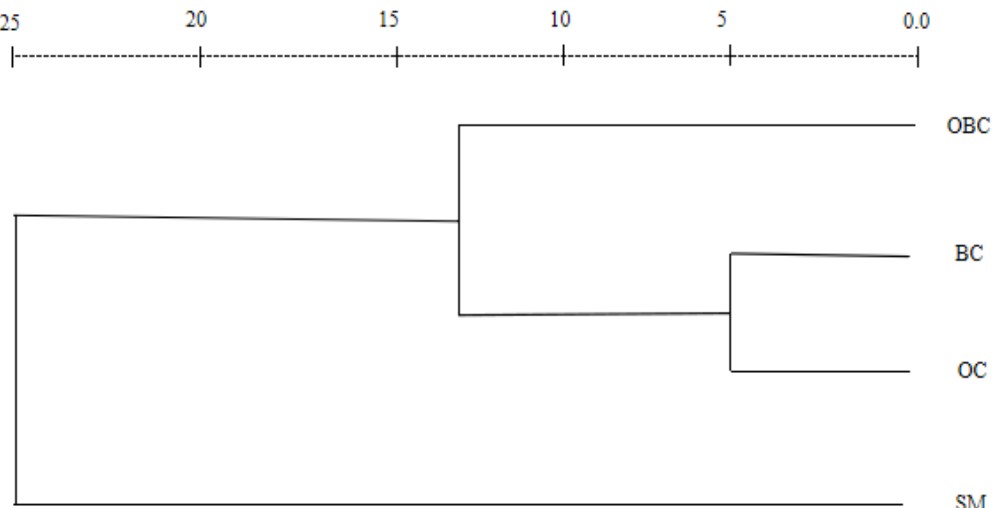

**Figure 2.** Dendrogram showing clusters from the four morphotypes of *M. rosenbergii.* SM = small male, BC = blue claw, OC = orange claw, OBC = old blue claw.

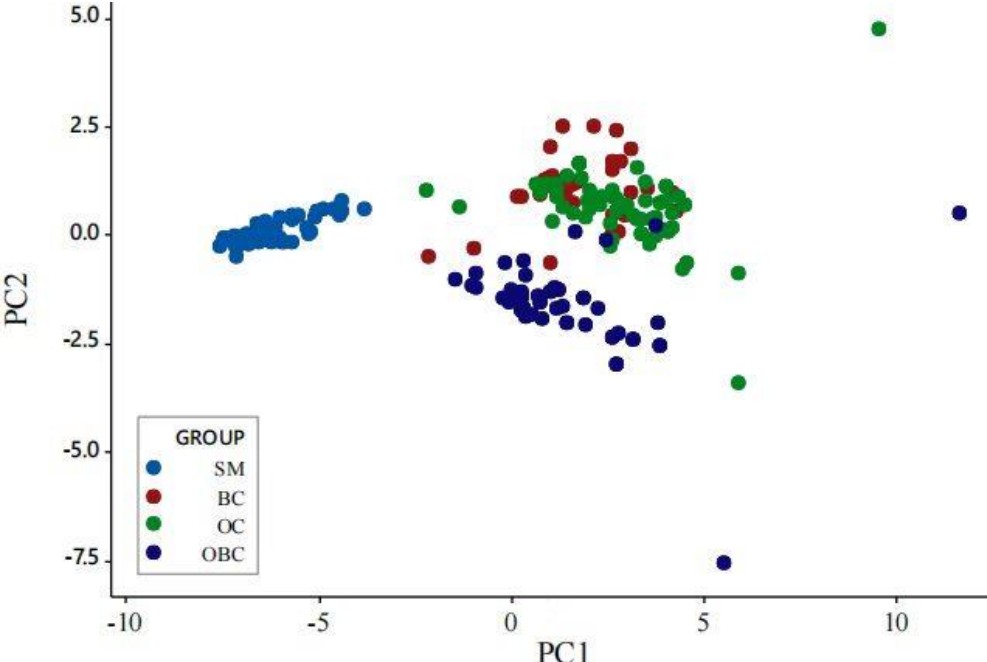

**Figure 3.** Scatter plot of the first and second principal components of morphometric traits for four male groups of *M. rosenbergii.* SM = small male, BC = blue claw, OC = orange claw, OBC = old blue claw.

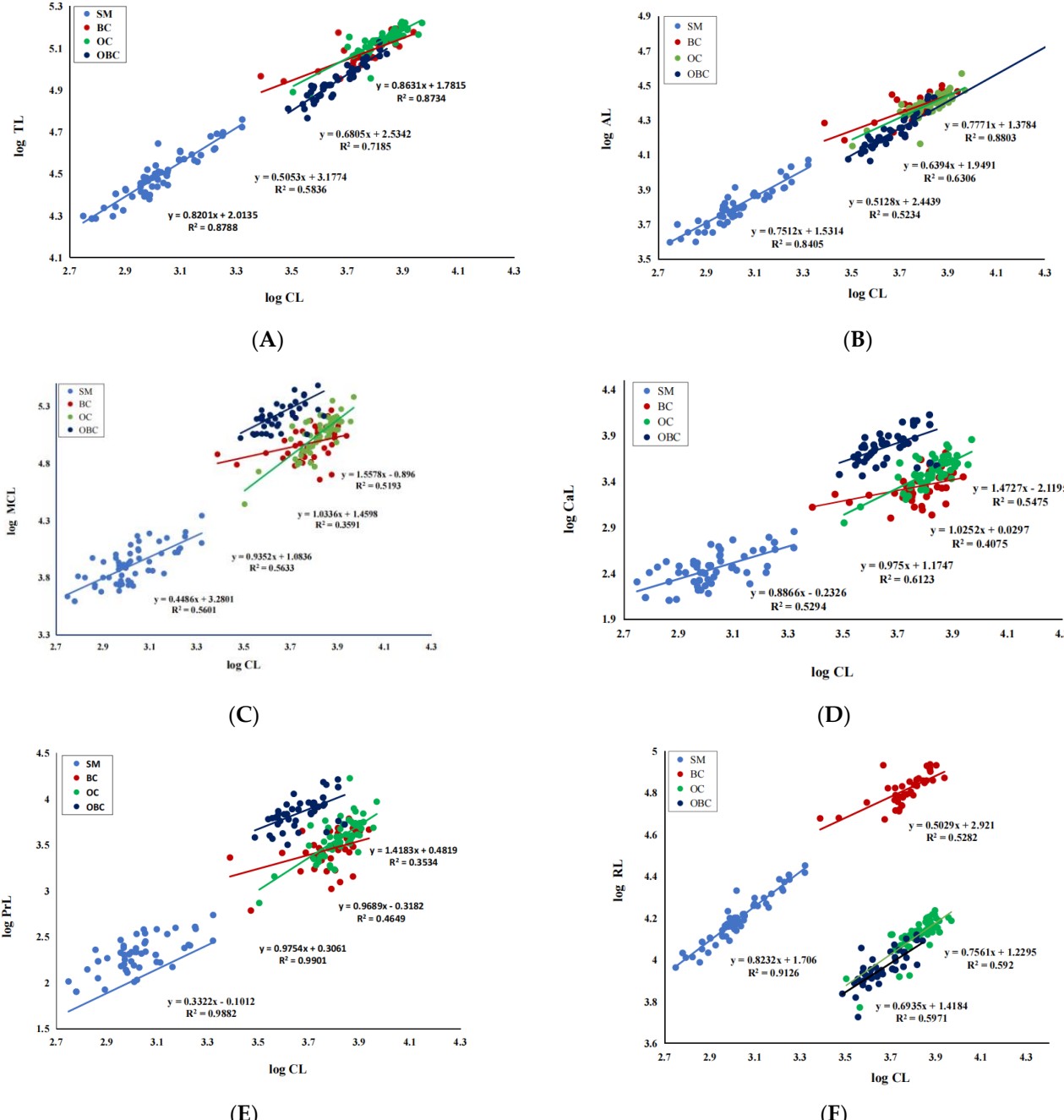

**Figure 4.** Scatterplots of seven morphometric variables against carapace length for *M. rosenbergii* male morphotypes. (**A**). TL vs. CL; (**B**). AL vs. CL; (**C**). MCL vs. CL; (**D**). CaL vs. CL; (**E**). PrL vs. CL; (**F**). RL vs. CL. CL = Carapace length, TL = Total length, AL = Abdominal length, MCL = Major cheliped length, CaL = Carpus length, PrL = Propodus length, and RL = Rostrum length.

**Table 3.** Regression analysis of morphometric data in the four male morphotypes of *M. rosenbergii*.

| Relationship | Morphotypes | n | a | b | r² | p-Value | Allometry |
|---|---|---|---|---|---|---|---|
| TL vs. CL | SM | 62 | 2.01 | 0.82 | 0.88 | 0.000 | − |
| | BC | 40 | 3.20 | 0.51 | 0.58 | 0.001 | − |
| | OC | 62 | 2.50 | 0.68 | 0.72 | 0.000 | − |
| | OBC | 51 | 1.78 | 0.86 | 0.87 | 0.0047 | − |
| AL vs. CL | SM | 62 | 1.53 | 0.75 | 0.84 | 0.000 | − |
| | BC | 40 | 2.44 | 0.51 | 0.52 | 0.000 | − |
| | OC | 62 | 1.95 | 0.64 | 0.63 | 0.000 | − |
| | OBC | 51 | 1.38 | 0.77 | 0.88 | 0.000 | − |
| MCL vs. CL | SM | 62 | 3.28 | 0.45 | 0.56 | 0.000 | − |
| | BC | 40 | 1.08 | 0.93 | 0.56 | 0.000 | = |
| | OC | 62 | 1.46 | 1.03 | 0.36 | 0.000 | = |
| | OBC | 51 | −0.89 | 1.55 | 0.52 | 0.075 | + |
| CaL vs. CL | SM | 62 | −0.23 | 0.88 | 0.53 | 0.000 | − |
| | BC | 40 | 1.17 | 0.97 | 0.61 | 0.193 | = |
| | OC | 62 | 0.03 | 1.02 | 0.41 | 0.000 | = |
| | OBC | 51 | −2.12 | 1.47 | 0.55 | 0.007 | + |
| PrL vs. CL | SM | 62 | 0.10 | 0.33 | 0.99 | 0.000 | − |
| | BC | 40 | 0.31 | 0.97 | 0.99 | 0.000 | = |
| | OC | 62 | 0.32 | 0.97 | 0.46 | 0.000 | = |
| | OBC | 51 | 0.48 | 1.42 | 0.35 | 0.000 | + |
| RL vs. CL | SM | 62 | 1.70 | 0.82 | 0.91 | 0.000 | − |
| | BC | 40 | 2.9 | 0.50 | 0.53 | 0.000 | − |
| | OC | 62 | 1.23 | 0.76 | 0.59 | 0.000 | − |
| | OBC | 51 | 1.42 | 0.69 | 0.60 | 0.000 | − |

Note: The carapace length (CL) was used as the independent variable ($x$) in the allometric equation $y = ax^b$, which was linearized by the equation $\log y = b \times \log x + \log a$, where $y$ is the length of a given body structure, $a$ is the intercept, and $b$ is the allometric coefficient. "−" means negatively allometric, "+" means positively allometric, and "=" means isometric. Positive allometry when $b > 1.1$, negative allometry when $b < 0.9$, and isometry at $0.9 < b < 1.1$.

### 3.3. Description of Major Cheliped in Different Male Morphotypes

The four morphotypes differed significantly, based on the chelipeds' color, length, and spination. Table 4 presents the details of the cheliped variables of each morphotype. Figures 5 and 6 illustrate the four male morphotypes of *M. rosenbergii* and their variation in cheliped morphology. The Turkey HSD multiple comparisons test revealed that spine height and angulation differed substantially among the four morphotypes (Table 4).

**Table 4.** Mean and range of values for spine angle and height of individual morphotypes in male *M. rosenbergii*.

| Morphotype | Segment | Spine Height | | | Spine Angle | | | Spination | Color |
|---|---|---|---|---|---|---|---|---|---|
| | | Mean | SD | Range | Mean | SD | Range | | |
| OC | dactylus | 0.52 | 0.09 | 0.36–0.65 | 60.25 | 6.05 | 51.52–73.8 | + | orange |
| | ischium | 0.43 | 0.05 | 0.36–0.50 | 64.31 | 3.67 | 52–67.4 | ++ | beige blue |
| | merus | 0.74 | 0.18 | 0.5–1.04 | 61.59 | 3.92 | 53.13–68.4 | + + | pale blue |
| | carpus | 0.81 | 0.07 | 0.68–0.94 | 59.37 | 5.42 | 42.3–67 | + + | blue |
| | propodus | 0.61 | 0.09 | 0.51–0.90 | 61.75 | 6.05 | 53–75.3 | + + | orange |

**Table 4.** *Cont.*

| Morphotype | Segment | Spine Height | | | Spine Angle | | | Spination | Color |
|---|---|---|---|---|---|---|---|---|---|
| | | **Mean** | **SD** | **Range** | **Mean** | **SD** | **Range** | | |
| BC | dactylus | 0.62 | 0.11 | 0.30–0.72 | 67.48 | 4.56 | 38.78–78.91 | + | deep blue |
| | ischium | 0.51 | 0.09 | 0.28–0.55 | 63.12 | 9.16 | 45–73.6 | + + | pale |
| | merus | 0.86 | 0.22 | 0.57–1.22 | 76.38 | 5.68 | 62.5–83.4 | + + | beige blue |
| | carpus | 1.07 | 0.2 | 0.73–1.41 | 83.00 | 7.64 | 64.8–92.5 | + + + | blue |
| | propodus | 0.67 | 0.11 | 0.45–0.97 | 68.99 | 4.56 | 58.7–77.5 | + + + | blue |
| OBC | dactylus | 0.71 | 0.11 | 0.56–1.00 | 60.70 | 10.05 | 37.2–77.41 | + | deep blue |
| | ischium | 0.48 | 0.08 | 0.3–0.61 | 54.66 | 7.99 | 40.6–70.1 | + + | light blue |
| | merus | 0.86 | 0.17 | 0.55–1.13 | 57.37 | 5.25 | 50–68.03 | + + + | deep blue |
| | carpus | 1.16 | 0.19 | 0.87–1.44 | 76.00 | 7.05 | 66.13–96.1 | + + + | deep blue |
| | propodus | 0.86 | 0.11 | 0.72–1.15 | 62.20 | 10.06 | 38.8–79 | + + + | deep blue |

Note: BC, blue claw; OC, orange claw; OBC, old blue claw. The small male morphotype has no spines. The unit for height and angle is mm and °, respectively. The plus sign (+) represents the presence and magnitude of spines on the chelipeds, and a higher number of plus signs means more spines.

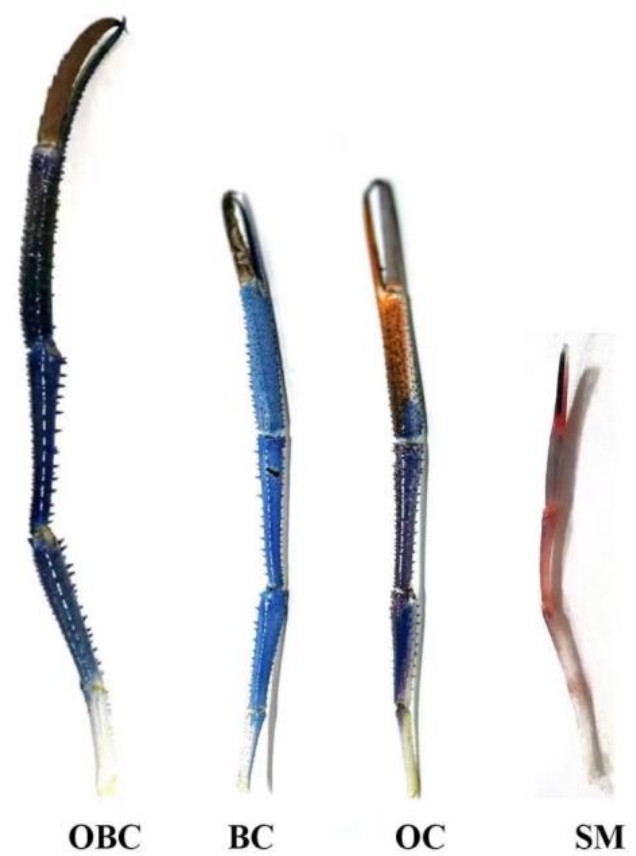

**Figure 5.** Detailed view of the traits of the major chelipeds of each male morphotype in *M. rosenbergii*. SM = small male, BC = blue claw, OC = orange claw, and OBC = old blue claw.

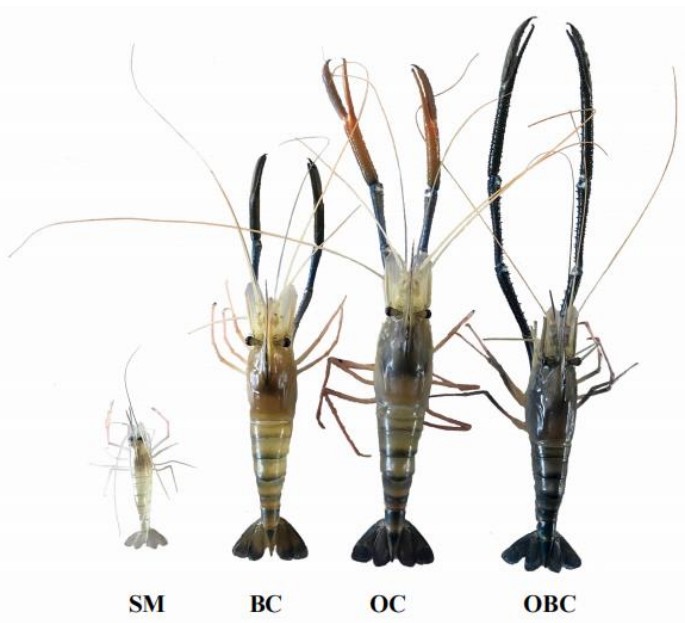

**Figure 6.** Four male morphotypes of *M. rosenbergii* captured in the cultivated population. SM = Small Male, BC = Blue Claw, OC = Orange Claw, and OBC = Old Blue Claw.

The major cheliped length (MCL) of small male prawns ranged from 36.40 to 76.89 mm (Table 1), translucent and devoid of spines or tubercles. In some prawns, the propodus and dactylus were pale orange, with pink pigmentation at the joints. No spines were recorded in all the segments of the SM group. The OC morphotype presented MCL ranging from 85.00 to 196.00 mm, with somewhat fewer spines than BC prawns, and the segments were opaque. Their ischium was opaque, slightly blueish, and had a few spines, although small tubercles often occurred. The spines had a mean height of $0.43 \pm 0.05$ mm and a mean angle of $64.31 \pm 3.67°$. Pale blue coloring was predominant in the merus and carpus and orange in the propodus and dactylus. The spines of the merus, carpus, and propodus were long and robust with mean height and angle of $0.74 \pm 0.18$ mm, $61.59 \pm 3.92°$; $0.81 \pm 0.07$ mm, $59.37 \pm 5.42°$; and $0.61 \pm 0.09$ mm, $61.75 \pm 6.05°$, respectively. The dactylus was orange and had a brownish-blue spot at the joint with the propodus, with widely spaced slender spines of mean height and angle of $0.52 \pm 0.09$ mm and $60.25 \pm 6.05°$, respectively.

The major cheliped length (MCL) in the BC morphotype ranged between 105.37 and 275.20 mm. The ischium was pale, with relatively longer spines ($0.51 \pm 0.09$ mm) and a larger spine angle ($63.12 \pm 9.16°$) than the OBC morphotype. The merus was beige-blue, and presented a few spines with a mean height of $0.86 \pm 0.22$ mm and a mean angle of $76.38 \pm 5.68°$. The carpus and the rest of the segments were blue. There were many spines on the carpus and propodus. The mean height was $1.07 \pm 0.20$ mm in the carpus, while the mean angle was $83.00 \pm 7.64°$. In the propodus, the mean height was $0.67 \pm 0.11$ mm, and the mean angle was $68.99 \pm 4.56°$. The dactylus presented as deep blue with a brownish spot at the joint with the propodus. The spine's mean height was $0.62 \pm 0.11$ mm, and the mean angle was $67.48 \pm 4.56°$.

In the OBC morphotype, the MCL range was 151.5–260 mm (Table 1), much larger than the other morphotypes. All segments were a similar blue color, and opaque. The carpus and propodus contained a sequence of long and well-developed spines that were uniformly distributed. In the carpus, the mean angle was $76.5 \pm 2.76°$, and the mean height was $1.1 \pm 0.19$ mm. In the propodus, the height was $0.86 \pm 0.11$ mm, and the mean angle was $62.2 \pm 10.1°$. The ischium had few well-developed spines with a mean height of $0.48 \pm 0.08$ mm and a mean angle of $54.66 \pm 7.99°$. The merus had several well-developed spines with a mean height of $0.86 \pm 0.17$ mm, a mean angle of $57.37 \pm 5.25°$, and a few tubercles in the dorsal region. The dactylus region was opaque, deep blue, and had slender spines with a mean height of $0.71 \pm 0.11$ mm, and a mean angle of $60.7 \pm 10.05°$.

In general, the spine set from the ischium to the propodus was more developed (long and robust) and more evenly spaced in the inner than the outer margin, with the animal in anatomical position (Figure 6). There was a concentration of simple setae in the dorsal portion of the dactylus region of the larger morphotypes (OC, BC, and OBC). The carapace of all groups was smooth, and devoid of spinules or tubercles, as seen in other *Macrobrachium* species.

## 4. Discussions

### 4.1. Morphotype Diversity

The study of the range and mean of various morphometric data of *M. rosenbergii* showed that the four groups represented a specific variation in morphometric characters, as demonstrated by ANOVA (Table 1). Our results revealed that the cultivated male population of *M. rosenbergii* from culture ponds in the Yangtze River delta, China, comprises four morphotypes of adult males differing in cheliped morphology, size, and morphometric relationships. *M. rosenbergii* have a variable number of morphotypes. It is believed that abiotic and population factors such as nutrient supply, temperature, and social growth control have a significant impact on the number of morphotypes in a population[23,24]. In *M. amazonicum*, multiple morphotypes are found in estuarine populations [25–29], while fewer morphotypes are from inland rivers [18,30,31], and, in some cases, only one, depending on local environmental characteristics [32,33].

The giant freshwater prawn, *M. rosenbergii,* has received the most research attention, due to its importance in aquaculture [34], placing this species among the earliest examples of studies on the heterogeneous growth of male prawns in decapod crustaceans. Although the exact cause for the size heterogeneity among male morphotypes is not understood, it is believed to be a cumulative effect of intrinsic and extrinsic factors [35]. The intrinsic variables are generally associated with biological and genetic differentiation, as well as inborn features connected with ontogenesis [36]. On the other hand, extrinsic factors such as environmental conditions, hierarchy position and prawn density, are thought to be the most important factors influencing differential growth [34,37]. Genetic characterization of *M. rosenbergii* male morphotypes shows that the morphotypes differ significantly at the molecular level, and morphotype differentiation processes are caused by variations in gene expression patterns among the male morphotypes [38,39]. Meanwhile, the genetic make-up of the population also affects the morphotypes, and higher inbreeding levels lead to the early development of smaller BC males [40,41]. Size variability of the harvested morphotype is crucial for overall profitability of GFP farming, because prawn market prices are highly dependent on individual size [38]. Recent studies of the growth patterns of *M. rosenbergii* male morphotypes showed a significant additive genetic component for male morphotypes [16,36], with prospects for genetic selection to change population structure in favor of the desired GFP male morphotypes. These developments can facilitate the production of larger prawns with a uniform weight.

There was a size overlap between OC and BC morphotypes, as the morphometric difference in most traits between OC and BC morphotypes was relatively insignificant (Table 1), suggesting little difference between the two morphotypes. In contrast, the morphotypes SM and OBC showed more differences from the others, as seen in the dendrograms and scatter plots (Figures 2 and 3). This is similar to *M. amazonicum* [7], where Green Claw 1 (GC1) is quite different from Cinnamon Claw (CC) and is similar to Green Claw 2 (GC2). In *Macrobrachium* species, different morphotypes can have similar body sizes but different cheliped sizes and ornamentation (color, spination pattern, presence of setae), resulting in distinct relative growth patterns [42]. In species where body size overlaps between morphotypes, differential patterns of chelipeds become the most important traits in the establishment of hierarchies between individuals within the same morphotype [42,43]. Similar to this study, a wide range of body-size overlap between morphotypes has been observed in *M. acanthurus* [42], *M. grandimanus* [5], and *M. amazonicum* [7].

This study observed the largest morphotype in the OC group, having recorded the highest mean number of almost all morphometric variables. This observation contradicts the normal biology of this species, since in an individual developmental pathway (from SM to OC to BC), OC males transform into BC and not vice versa. Hence, true BC should be larger, on average. However, at the population level, this may be possible when considering the average size of each morphotype, as each male transforms into the next morphotype at different times/sizes, so in some populations, one can find many large OCs and only a small part of the BC group that are large, which is true in this case, since the largest individual was found in the BC group.

Other useful characteristics that contribute to the differentiation among male morphotypes of *M. rosenbergii* include telson length (TeL), uropod length (UL), and rostrum length (RL). The length of the telson and the uropod are proportionally longer in BC males than in other morphotypes. Observations from this study showed a significant variation ($p < 0.05$) in the rostrum length (Table 1). Differences in the rostrum morphology do occur within populations of the *Macrobrachium* species. Thus, rostrum characteristics may be essential in classifying prawns into morphotypes [8].

### 4.2. Relative Growth Patterns

Several studies have been made on the relative growth patterns of *M. rosenbergii* morphometric traits to explain the morphological distinction among adult male morphotypes [15,17,23]. According to the allometric growth constant obtained in the morphometric analysis in this study, each group presents a specific growth pattern of body relationships, indicating that the four male morphotypes have different growth rates. Most relationships showed negative allometry in SM, isometry in BC and OC, and positive allometry in OBC. Among the morphometric relationships used to describe the relative growth of *M. rosenbergii*, the TL/AL/RL vs. CL relationships presented a similar pattern, with good coefficients of determination, and were the equations that best described the relative growth of this species.

Based on the different growth rates (Table 3), we inferred that *M. rosenbergii* male morphotypes had undergone rapid growth in their developmental pathway [17], as seen in *M. amazonicum* [7] and *M. brasiliense* [10]. For the social hierarchy, the BC males are dominant, followed by OC males, and the SM males are in the lowest position. OBC males are senescent individuals and are believed to evolve from BC males, with relatively small body sizes in carapace length and major cheliped length [15]. OC males have been reported to transform into the BC morphotype only when the largest OC individual becomes larger than the largest BC in their physical vicinity [44]. Once an OC male transforms into a BC male, the rapid growth that characterizes the OC morphotype ceases. Subsequently, the new BC individual inhibits the growth of subordinate individuals of the same age class [38]. Orange claws are the fastest growing of all the male morphotypes [34], and their proportion in cultured populations influences the productivity of GFP.

The change from one morphotype to the next can happen in a single molt or a gradual process, and retrocession may occur [7]. However, Karplus et al. (2000) [6] confirmed an obligatory sequence in the development of *M. rosenbergii*. According to available literature, the transition from SM to OC is gradual, whereas the change from OC to BC is abrupt and happens in a single metamorphic molt, resulting in evident changes in the cheliped morphology (coloration and spination) [17]. On the other hand, changes between morphotypes in other decapod species are less distinct [45,46], and identifying specific morphotypes requires multiple criteria [11].

### 4.3. Morphological Diversity of Chelipeds

The wide difference in the range observed in the major cheliped amongst the morphotypes indicates its usefulness for identifying the phenotypic differences in the prawn population. *M. rosenbergii* morphotypes are clearly distinguished, based on the morphological variation of the major cheliped. Allometric growth of chelipeds has been extensively

studied in adult males of the GFP [15,17,23]. In this study, the growth of the major chelipeds (MCL), carpus (CaL), and propodus (PrL) showed the same allometric growth pattern in all four groups. These structures are therefore developing at the same rhythm across groups. The growth of these structures was negatively allometric in SM males, isometric in BC and OC males, and positively allometric in OBC. This growth pattern of the cheliped structures described here conforms well with related species from the *Macrobrachium* genus, where male chelipeds, particularly the carpus and propodus, exhibit marked allometric growth ([11,47]. However, the best morphometric discriminator between BC and OC was the relationship between carpus length and carapace length, followed by the relationship between propodus length and carapace length. This is supported by the findings of Kuris et al. (1987) [17]. Furthermore, the chelipeds' spination pattern differed significantly among the four morphotypes of our specimens, as recorded in *M. amazonicum* [7,18] and *M. Brasiliense* [10]. The small male (SM) morphotypes presented smaller chelipeds, devoid of spines, as compared with the other morphotypes: orange claw (OC), blue claw (BC), and old blue claw (OBC) [17]. The OBC morphotype was characterized by proportionally longer and more robust spines, whose orientation differs from the BC and OC morphotypes.

Morphological variations of the major chelipeds have been established in many commercial crustacean species, mainly in males (e.g., [46–48]). So far, the few identified morphotypes out of about 250 species of the *Macrobrachium* genus all differ in cheliped morphology [5,7,9,10,49]. According to these authors, chelipeds are one of the morphometric traits that can be highly varied among the morphotypes of *Macrobrachium*. Thus, differences in the chelipeds' morphology are essential in classifying the prawns into morphotypes. Differential cheliped patterns, particularly in the propodus and dactylus, have been used to distinguish male morphotypes of *Macrobrachium* [42]. According to Hoshan et al. (2022) [49], small differences in cheliped measurements can reveal the dominant individuals in a population, that is, individuals of the higher cast become dominant because they have more developed chelipeds, regardless of their body size. Studies on various species of *Macrobrachium* emphasized the importance of chelipeds for predation, aggression, mating, and protection of females by males [10,13,50]. Males with well-developed chelipeds have greater reproductive success. They can explore a large area for food and prey without having to move to allow the occupation of territory and maintenance of social structure [5]. The size of the major cheliped is often related to the morphotype and social status of male *M. rosenbergii*.

In *Macrobrachium*, each male morphotype invests a different amount of energy in developing the major cheliped (or its segments) throughout the life cycle [7], which changes proportionally with the respective functions at each phase. The morphotype with large chelipeds invests relatively more energy in developing the chelipeds, at the expense of body growth [7,10]. This could explain why the OBC males' bodies do not develop as much as in other larger morphotypes. The OBC is a special male individual, with a major cheliped more developed than the normal BC type [51].

The characteristics of each morphotype's chelipeds impart a distinct role in the population and the environment in which it lives. The differential pattern in cheliped size, color, and spination certainly has an impact on the intraspecific interactions as well as the male's interaction with the environment [5,51]. The use of spines to differentiate male morphotypes is common in *Macrobrachium* species [5,10], as well as in other genera, such as *Rhynchocinetes* [45,46]. Spines are very important defensive structures for prawns. The number and morphology of the spines are closely related to the establishment of the hierarchy [52]. In the present study, individuals from the OBC morphotype have proportionally longer spines than those from the BC and OC groups, indicating that OBC might have prior position in relation to the other three groups.

The existence of morphotype differences directly influences the growth of undersized smaller prawns, via social dominance [13]. This phenomenon poses a massive challenge under the various practical production systems of GFP farming. Knowledge of the population biology of *M. rosenbergii*, the extent of variation, and the relationship among all male

morphotypes contributes significantly to our understanding of the social biology of the species and the optimization of culture management.

## 5. Conclusions

Overall, the present results revealed that the GFP male populations in the culture ponds of the Yangtze River delta, China, comprised four morphotypes instead of the three basic morphotypes reported earlier. These results must also be confirmed through genetic techniques. Further studies are needed to determine whether there is a correspondence between each morphotype's physiological, behavioral, and functional characteristics. A clear definition of morphotypes of this economic species is extremely important for understanding its growth processes and adaptive value in the population, to optimize culture management.

**Author Contributions:** S.I., conceptualization, data curation, methodology, writing original draft; Z.Z. and J.L., methodology, data curation; X.L., software, investigation; Q.T., funding acquisition, writing—review and editing; Z.X., sources, investigation; S.Y., software, validation; G.Y., supervision, funding acquisition. All authors contributed to the article and approved the submitted version. All authors have read and agreed to the published version of the manuscript.

**Funding:** This study was supported by the Basic Public Welfare Research Project of Zhejiang Province (LGN20C190007), the National Key R&D Programme of China for Blue Granary (2018YFD0901300), the Major Research & Development Programme (Modern Agriculture) of Jiangsu Province (BE2019352), and the Earmarked Fund (CARS-48).

**Institutional Review Board Statement:** This work was carried out in accordance with the Ministry of Science and Technology's Guide for Laboratory Animals (Beijing, China). Huzhou University authorized the experimental protocol for animal care and tissue collection (ethical approval code 20190625).

**Informed Consent Statement:** Not applicable.

**Data Availability Statement:** All supporting data are included within the main article.

**Acknowledgments:** We thank Jingfen Li, Miaoying Cai, Haihu Tu and Xin Peng for their help in sample and data collection. Thanks are also given to two anonymous reviewers and editors for their valuable advice on revising our manuscript.

**Conflicts of Interest:** The authors declare no conflict of interest.

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
