# Peer review of "Morphological Diversity of Different Male Morphotypes of Giant Freshwater Prawn Macrobrachium rosenbergii (De Man, 1879)"

_2673-9496, doi:10.3390/aquacj3020012_

Round 1
Reviewer 1 Report
Review of ‘Morphological diversity of different male morphotypes of giant freshwater prawn Macrobrachium rosenbergii (De Man, 1879)’ by Salifu Ibrahim, Zhenxiao Zhong, Xuan Lan, Haihui Tu, Xin Peng, Jinping Luo, Qiongying Tang, Zhen-glong Xia, Miaoying Cai, Shaokui Yi, Jingfen Li, and Guoliang Yang.
The authors conducted a study aimed to describe the morphological characteristics of reared male giant freshwater prawn Macrobrachium rosenbergii and distinguish morphotypes. They measured and weighed prawn specimens and compared the data. The authors found 4 morphotypes and concluded that these exhibit different growth patterns. The authors discussed their results and compared them with previous reports. However, additional comparisons are required to clarify the conclusions.
The authors should modify the presentation of citations according to the Rules for Authors.
Abstract
Introduction
Pg 2 Ln 67: Suggest changing ‘Aside the three most’ to ‘Aside from the three most’
Pg 2 Ln 71: Suggest changing ‘we identified and added the morphotype’ to ‘we identified and added a new morphotype’
Methods
Pg 2 Ln 85: Suggest changing ‘140-days growth’ to ‘140 days of growth’
Pg 4 Ln 129: Suggest changing ‘data was subjected to Shapiro–Wilk’ to ‘data were subjected to the Shapiro–Wilk’
Results
The authors did not compare statistically the four groups. Both PCA and cluster analysis provide ordination and separation between groups, but additional tests are required to provide p-values and confirm significant differences, for example, ANOSIM or PERMANOVA test followed by pair-wise comparisons.
Pg 8, Fig. 2: Please, add the legend to the OX axis.
Discussion
Pg 17 Ln 466: Suggest changing ‘food and preys’ to ‘food and prey’
The authors should update the discussion by some recent citations:
Rios, D. P., Pantaleão, J.A.F, Hirose, G.L. (2021). Occurrence of male morphotypes in the freshwater prawn Macrobrachium acanthurus Wiegmann, 1836 (Decapoda, Palaemonidae). Invertebrate Reproduction and Development, 65(4), 268-278.
Hoshan, I., Yesmin, A., Ray, M., Ahmed, F. F., Mahfuj, S. (2022). Multivariate morphometric differentiation of Macrobrachium species (Crustacea: Palaemonidae) along the northern rivers of Bangladesh. Bangladesh Journal of Fisheries, 34(1), 27–39.
Rossi, N., Pantaleão, J. A. F., Mantelatto, F. L. (2022). Integrated morphometric and molecular analyses indicate three male morphotypes in the freshwater prawn Macrobrachium olfersii (Decapoda, Palaemonidae) along the Brazilian neotropical region. Acta Zoologica. https://doi.org/10.1111/azo.12437
Conclusion
This statement seems to be confusing "Overall, the analysis developed in this study supported the separation of the M. rosenbergii males into four distinct morphotypes, also showing morphometric similarity in the BC and OC" and must be supported by statistical comparisons.
From the data presented, the reader can see 3 morphotypes, because there is a significant overlap of points representing BC and OC in Fig. 3 and these groups belong to one cluster (Fig. 2).
Author Response
Response to reviewer 1 comments

Reviewer 2 Report
The manuscript of the titled “Morphological diversity of different male morphotypes of giant freshwater prawn Macrobrachium rosenbergii (De Man, 1879)” showed four different morphotypes for male population of the prawn in China. It looks that there is large phenotypic variations among the male population of prawn. If it is so, I suggest the author redesign the experiment to investigate the genetic basis of the phenotypic variation that could attracted more interests from the audience. Besides, how to determine the “Small Male” that is adult? I guess it may be the younger one. Therefore, it may be not suitable to compare the phenotypes of the four kind males (SM, BC, OC and OBC) of prawn. Besides, the expression of some language is not very clear.
The other minor issues:
1. Line71, “We identified and added the morphotype, old blue claw males (OBC), to the three most discussed male morphotypes (small males (SM), blue claw (BC), orange claw (OC)), thus using four morphotypes as our experimental materials to analyze the extent of variability and relationship”. However,Line85,“Male prawns were collected from a single-age population of about 140-days growth”. It shows that the four morphotypes experimental materials are of the same age. Please describe in detail how to distinguish and obtain old blue claw and blue claw.
2. The text description in the result part is inconsistent with the Table 1 data. The author is requested to carefully check the consistency between the text description and the experimental data chart.
For example, Line165, “All observed morphometric characteristics demonstrated a highly significant difference (P < 0.05) among the four morphotypes except for rostrum length (RL), abdominal length (AL), carpus width (CaW), propodus width (PrW), carapace width (CW) between OC and BC, and abdominal width (AW), abdominal depth (AD), carapace depth (CD) between BC, OC and OBC”.
According to the data in Table 1, there is a significant difference between OC (61.66±4.69a) and BC (59.42±5.19b) in rostrum length (RL). However, the author described it as having no significant differences in the text description of the results.
In addition, according to the data in Table 1, there is no significant difference between OC (21.23±4.84a) and BC (20.08±2.30ab) in Abdominal width (AW). However, the author described it as having significant differences in the text description of the results.
In addition, according to the data in Table 1, there is no significant difference between OC (21.23±4.84a) and BC (20.08±2.30ab) and between OC (21.23±4.84a) and OBC (18.33±5.24b) in Abdominal width (AW). But, there is a significant difference between OC (21.23±4.84a) and OBC (18.33±5.24b). However, the author described it as having no significant differences between BC, OC and OBC in the text description of the results.
3. The text description in the result part is inconsistent with the Table 2 data.
Line173, “According to the loadings of component coefficients obtained for morphometric data, the most influential variables for PC1 included TL, RL, BL, CL, CaW, CD, AL, AD, CW, and AW”. According to the data in Table 2, in addition to these ten morphological traits (TL, RL, BL, CL, CaW, CD, AL, AD, CW, and AW.), Cw (Carapace weight) should also be included.
4. Line 234, “Linear regressions showed that TL vs. CL, AL vs. CL, and RL vs. CL relationships were negatively allometric in all groups”. However, neither Figure 4 nor Table3 clearly shows the negative allometric growth value. In addition, please specify how to determine the type of allometry (positively allometric, negatively allometric, isometric).
5. There is an extra line of space in Table 3.
6. There are some writing errors in Table 4. For example,“++” and “beige-blue”.
7. Line 303, “The ischium was pale, with relatively shorter spines (0.51 ± 0.09 mm) and a smaller spine angle (63.12 ± 9.16 °)”. “shorter” and “smaller”, please explain the conclusion drawn by comparing with which morphotypes. According to the data in Table 4, the ischium of BC (0.51mm) is larger than OC (0.43mm) and OBC (0.48mm).
8. The format of references is not uniform. For example, Line 525 “Aquaculture” “pp.168 184”, Line 612 “39: 55 – 64.”, Line 631 “51 (12), 5040-5049”, Line 645 “(2020)” should be consistent with other reference formats!
9. Some of the literatures are very old, with 20 literatures before 2000, accounting for nearly one third. Please cite the latest research as much as possible.
Author Response
Response to reviewer 2 comments

Round 2
Reviewer 1 Report
Please, consider the following:
L 92. " The OBC morphotype is characterised by relatively smaller abdominal length in carapace
length…" This is unclear. Did you mean "abdominal length in relation to carapace length"?
Citations should be prepared according to the requirements of the journal.
Author Response
Response to Reviewer 1 Comments

Reviewer 2 Report
The author had completed the revision according to the comments. I think the revision version can be accepted.
Author Response
Response to Reviewer 2 Comments
